# Brief Communication: Accurate and autonomous snow water equivalent measurements using a cosmic ray sensor on a Himalayan glacier

Navaraj Pokhrel[1,2], Patrick Wagnon[1], Fanny Brun[1], Arbindra Khadka[1,2,3], Tom Matthews[4], Audrey Goutard[1], Dibas Shrestha[2], Baker Perry[5], Marion Réveillet[1]

[1]Univ. Grenoble Alpes, IRD, CNRS, INRAE, Grenoble INP, IGE, Grenoble, France
[2]Central Department of Hydrology and Meteorology, Tribhuvan University, Kirtipur, Nepal
[3]International Centre for Integrated Mountain Development, Kathmandu, Nepal
[4]Department of Geography, King's College London, London, United Kingdom
[5]Department of Geography and Planning, Appalachian State University, Boone, North Carolina, USA

*Correspondence to*: Navaraj Pokhrel (Navaraj.77558@cdhm.tu.edu.np)

**Abstract.** We analyze snow water equivalent (SWE) measurements from a cosmic ray sensor (CRS) on the lower accumulation area of Mera Glacier (Central Himalaya, Nepal) between November 2019 and November 2021. The CRS aligned well with field observations and revealed accumulation in pre-monsoon and monsoon, followed by ablation in post-monsoon and winter. COSIPY simulations suggest significant surface melting, water percolation and refreezing within the snowpack, consistent with CRS observations, yet liable to be missed by surface mass balance surveys. We conclude that CRS can be used to determine mass fluxes in various climatic settings, but the interpretation of the total changes in SWE needs complementary measurements and model analysis to determine the share of specific mass fluxes, such as melt and refreezing.

## 1 Introduction

Seasonal snowpack in high mountain regions is crucial for glaciology, hydrology and climate change research (Stewart, 2009). Snow accumulation and more generally high-altitude precipitation are major unknowns of the water cycle in the higher Himalayas (> 5000 m a.s.l.) (Immerzeel et al., 2015). Moreover, the quantitative assessment of snow water equivalent (SWE) in high-altitude snowpacks is vital for mitigating disaster risks, especially for floods and avalanches. In the logistically challenging environment of the higher Himalayas, accurately measuring precipitation and addressing snow distribution are difficult tasks, which limit the reliability and continuity of SWE measurements (Shea et al., 2015). The observational difficulties are compounded by high spatial heterogeneity in precipitation, driven by the complex interactions between topography and atmospheric circulation (Perry et al., 2020). There is hence a high incentive to overcome the logistical challenges and increase the density and quality of continuous SWE measurements.

A variety of techniques ranging from on-site measurements like snow pillows, snow pits, ground-penetrating radar to remote sensing tools like passive microwave and synthetic aperture radar, can be used to gather SWE data in remote areas, despite certain limitations and biases (Leinss et al., 2015). Kodama et al. (1975) and Kodama and Nakai (1979) introduced a method using cosmic ray neutrons. These neutrons, created by cosmic rays hitting Earth's atmosphere, interact with hydrogen in water. The decrease in neutron counts upon absorption allows for the estimation of SWE values (e.g., Gugerli et al., 2019). Measurements by cosmic ray sensors (CRS) offer autonomous, point measurements of SWE and therefore hold promise for greatly increasing the density of SWE observations in remote mountain regions.

In this study, we deployed a Hydroinnova SnowFox: a CRS able to measure SWE up to 2 meters water equivalent or more (Howat et al., 2018; Gugerli et al., 2019), in the lower part of the accumulation zone of Mera Glacier (Central Himalaya, upper Dudh Koshi basin). Since 2007, Mera Glacier has undergone systematic monitoring, establishing itself as one of the longest-running field-based series concerning mass and energy balance in the Himalayas. By using field measurements, in-situ meteorological data and adjacent weather stations, alongside a surface mass and energy balance model (COSIPY), our objectives are to: (i) analyze the SnowFox performance by comparing its SWE estimating with manual field observation; (ii) understand the seasonal evolution of snowpack in the accumulation zone of Mera Glacier; and (iii) utilize COSIPY to explain the processes driving snowpack evolution within Mera Glacier's accumulation zone.

## 2 Study area and climate setting

Mera (27.7° N; 86.9° E; 4.84 km$^2$ in 2018) is a debris-free glacier in eastern Nepal's upper Dudh Koshi basin, accumulating in summer (Wagnon et al., 2021). The SnowFox was installed on a large, flat area in the lower part of Mera Glacier's accumulation zone at 5770 m a.s.l., facing north and surrounded by crevasses (Fig. 1c).

Following (Bonasoni et al., 2010), we divided the year into four distinct seasons: winter (Dec-Feb) with colder, drier, windy conditions; pre-monsoon (Mar-May) with gradually increasing temperature and humidity, and less wind; monsoon (Jun-Sep) with light wind, constantly high temperature, and heavy precipitation due to moisture influx from the Bay of Bengal; post-monsoon (Oct-Nov) with drier, sunny, colder, windier weather, and occasional typhoons causing substantial snowfall above approximately 4000 m a.s.l. within a few days, (e.g., 18-20 October 2021) (Adhikari et al., 2024). In general, vertical, and horizontal gradients of precipitation are large due to complex topography over the region (Sherpa et al., 2017). However, this spatial variability is not well quantified due to the limited number of weather stations and inter-annual variability in total precipitation.

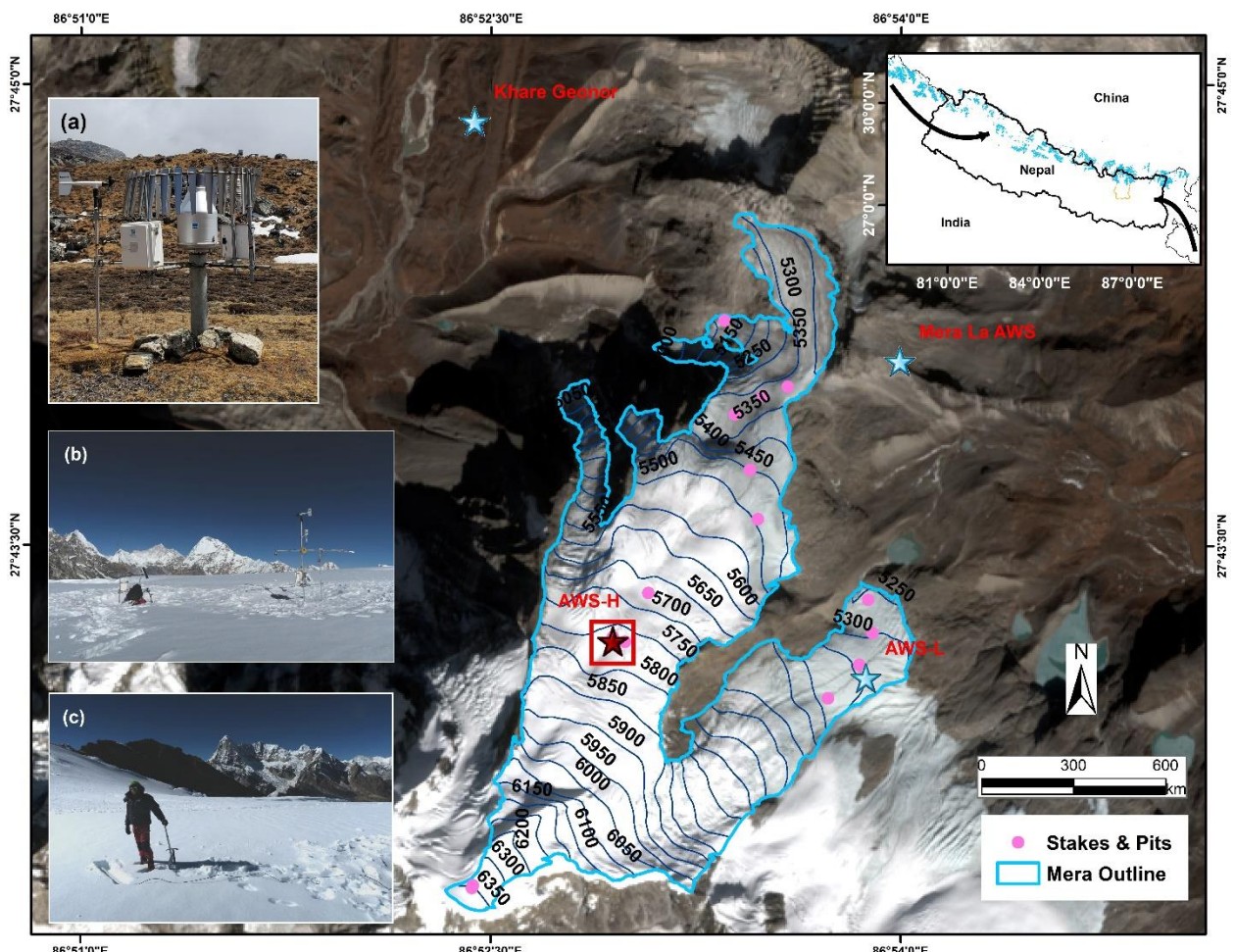

**Figure 1. Map of Mera Glacier showing the network of ablation stakes and accumulation pits (pink dots). The stars represent the locations of the AWS used in this study, with in red the location of the SnowFox installed approximately 10 m northwest of AWS-H.**
**The red square represents the grid cell at AWS-H where COSIPY simulations are done. The pictures show the all-weather Geonor precipitation gauge (a), a picture of the SnowFox site showing AWS-H on the right, the SnowFox datalogger mast on the left and the snow trench where the SnowFox is installed (in the front) (b), and the SnowFox during the installation on 12 November 2019 (c). The outline of Mera Glacier is from 2018 with a total area of 4.84 km², and the background image corresponds to Copernicus Sentinel data (Sentinel-2 image from 24 November 2018). Elevation contours are extracted from the 2012 Pléiades DEM (Wagnon**
**et al., 2021). The inset map gives the location of Dudh Koshi basin (yellow) in Nepal with general pathways of moisture carried by westerlies (left) and Indian summer monsoon (right). Light blue areas are the glacierized areas from Randolph glacier inventory 6 (Li et al., 2021).**

## 3 Data and Methods

In this study, we primarily utilize data from 12 November 2019 to 21 November 2021 of two on-glacier Automated Weather
Stations (AWS), namely AWS-H (equipped with SnowFox) in the accumulation area and AWS-L in the ablation area (Fig. 1). We also use data from AWS Mera-La, located off-glacier on solid rocks to fill the data gaps of the other AWSs. Additionally, the Khare Geonor station records precipitation in all weather conditions (Fig. 1b). This Geonor T-200B is equipped with a

wind shield and we apply undercatch corrections, as recommended by the World Meteorological Organization (WMO), more details are available in Khadka et al. (2024). In-situ measurement of snow density and snow depth were carried out on 24 November 2020 and 19 November 2021 using snow coring with manual drilling (Wagnon et al., 2021).

On 12 November 2019, the SnowFox was installed approximately 20 cm below the fresh snow surface, buried at 20 cm to be as shallow as possible, while avoiding exposure due to the wind deflation that is commonly observed in the post monsoon and winter. It was progressively buried by snowfalls, until we excavated it in November 2021, roughly two months after it stopped functioning due to burial its solar panel. The sensor counts downward falling secondary cosmic neutrons that pass through the snowpack, which is converted to SWE through a calibration function. In this case, the scanning time was one minute, and the records were averaged at hourly resolution.

The corrected neutron count for time step $i$ ($N_i$) is computed from the raw neutron count rate ($N_{raw,i}$) by accounting for variations in solar activity $(F_{s,i})$ and, more importantly, for changes in in-situ air pressure $(F_{p,i})$ (Gugerli et al., 2019); (Howat et al., 2018); (Jitnikovitch et al., 2021):

$$N_i = N_{raw,i}.F_{s,i}.F_{p,i} \tag{1}$$

Variations in $F_{s,i}$ are quantified with the aid of a (snow free) reference station. As there were not any neutron monitoring stations close to the study area, we used the solar correction factor established through the examination of data from 94 globally distributed, quality-controlled neutron counts station, which were found to be strongly correlated between the relative counting rates at the site of interest (Desilets, 2021) and at a reference neutron monitor center, the Jungfraujoch station (Flückiger and Bütikofer, 2009):

$$F_{s,i} = \left(1 - \frac{M_0}{M_{(t)}}(1 - \tau)\right) \tag{2}$$

where, $M_{(t)}$ is the neutron count of Jungfraujoch at time $t$, and $M_o$ is the counting rate at an arbitrary chosen reference time. $\tau$ is a dimensionless slope parameter adjusting the ratio to the site of interest. Its value depends on the effective cutoff rigidity ($R_c$) and atmospheric depth ($\chi$), both of which respectively depend on the altitude and latitude of the site of interest ((McJannet and Desilets, 2023)):

$$\tau(\chi, R_C) = \epsilon K(c_0 + c_1\chi)[1 - \exp(-[c_2 + c_3\chi]R_C{}^{c_4+c_5\chi})] \tag{3}$$

where, K represents a location-specific normalization factor, with a value of 3.08. $\epsilon$, which equals 1.14, is a correction factor used to adjust the sensitivity of the standard lead neutron monitor. Rc, set at 14.53 GV, refers to the effective vertical rigidity, calculated using the MAGNETOCOSMICS code (part of the Geant4 toolkit, available at crnslab.org). Lastly, $\chi$, equal to 543.51

105 g cm$^{-2}$, denotes the atmospheric depth, which is derived from local atmospheric pressure (Pa) and the acceleration due to gravity (g = 9.78 m s$^{-2}$):

$$\chi = \frac{10P_a}{g} \qquad (4)$$

Similarly, the following barometric pressure coefficients $c_0$ (7.977 $\times$ 10$^4$), $c_1$ (1.626), $c_2$ (3.990 $\times$ 10$^{-03}$), $c_3$ (5.476), $c_4$ (-1.527 $\times$ 10$^4$), $c_5$ (1.250) are used respectively (Desilets, 2021). In our application, we therefore find that $\tau$ has a value of 0.324.

Air pressure is not measured directly at the study site, due to pressure sensor failure; instead it is reconstructed based on the
110 Mera La station using the hydrostatic equation (Wallace and Hobbs, 2006). Finally, the pressure correction factor $F_{p,i}$ is obtained by:

$$F_{p,i} = exp\left(\frac{P_i - P_0}{L}\right) \qquad (5)$$

The observed hourly air pressure values are represented by $P_i$ while $P_0$ stands for a reference pressure. For the reference period, we chose a long-term (12 November 2019 to 21 September 2021) mean pressure value (531.6 hPa), used also in equation 4. The mass attenuation length (L) is taken as 150 g cm$^{-2}$ for our study site, this value was obtained from the online calculator
of the crnslab.org, and more specifically using the "Scaling factor calculator", which is an implementation of Mcjannet & Desilets (2022) and takes the latitude, longitude and elevation as inputs.

To calculate SWE, we used the relative neutron count ($N_{rel,i}$) i.e., the corrected neutron count ($N_i$) divided by a reference count ($N_o=150$) which is the averaged raw neutron count calculated from 1-minute means over 1-s time interval, between 11:00 and
120 12:00, local time, on 19 November 2021 while the sensor was running on-site, at the surface, after being excavated the day before. During this 1-hr run, the weather was cold and overcast, with very light snowfall. The relative neutron count is then used to derive SWE with the nonlinear equation:

$$SWE_i = -\frac{1}{\Lambda_i}.ln\ N_{rel,i} \qquad (6)$$

The variable $\Lambda_i$ is the effective attenuation length (in cm) with empirical values representative of a 'glacier landscape' ($\Lambda_{min}$ = 14 cm, $\Lambda_{max}$ = 114.1 cm, $a_1$ = 0.35, $a_2$ = 0.08 and $a_3$ = 1.117;(Jitnikovitch et al., 2021); (Howat et al., 2018):

$$\Lambda_i = \frac{1}{\Lambda_{max}} + \left(\frac{1}{\Lambda_{min}} - \frac{1}{\Lambda_{max}}\right).\left(1 + exp\left(\frac{a_1 - N_{rel,i}}{a_2}\right)\right)^{-a_3} \qquad (7)$$

SWE measurements are compared with COSIPY mass flux simulations. COSIPY is a one-dimensional multi-layer python-based model that resolves the energy and mass exchanges of a snowpack/ice column with the atmosphere (Sauter et al., 2020). The COSIPY simulation were produced by Khadka et al. (2024). COSIPY was forced with distributed meteorological forcings originating from an AWS located on the lower part of Mera glacier (Khadka et al., 2024). We extracted the mass fluxes modelled by COSIPY from the 0.003°×0.003° (0.01 km$^2$) grid cell where the SnowFox and AWS-H are located (Fig. 1). These

COPSIPY simulations were forced by in-situ atmospheric measurements recorded at AWS-L and have been previously validated against all available field measurements including albedo, surface temperature and point mass balances at the AWS-H site. The input precipitation is taken from Khare station without applying any altitudinal gradient.

## 4 Results and discussion

The SnowFox demonstrated its effectiveness in accurately determining SWE using cosmic rays through good agreement with

135 field measurements, even though we could assess only the cumulative SWE overs two separate years (Fig. 2a). During the sensor's installation, the SWE corresponding to 24 mm water equivalent (w.e.) from about 20 cm of fresh snow, translating to a snow density of 120 kg m$^{-3}$. On 24 November 2020, the sensor recorded a SWE of 562 mm w.e., closely matching the manually measured SWE of 533 $\pm$ 49 mm w.e., performed a few meters from the SnowFox location. The sensor was non-functional between 3 and 24 November 2020, because the solar panel was buried by snow; and it ceased functioning on 16

September 2021, for the same reason. At that time, the cumulative SWE measured by the sensor was 1282 mm w.e. The sensor was excavated on 18 November 2021, and manually measured SWE was equal to 1357 $\pm$ 88 mm w.e. During the first operational gap, 6 mm w.e. of precipitation was recorded at the Khare Geonor station, while 137 mm w.e. was recorded during the second gap of 63 days.

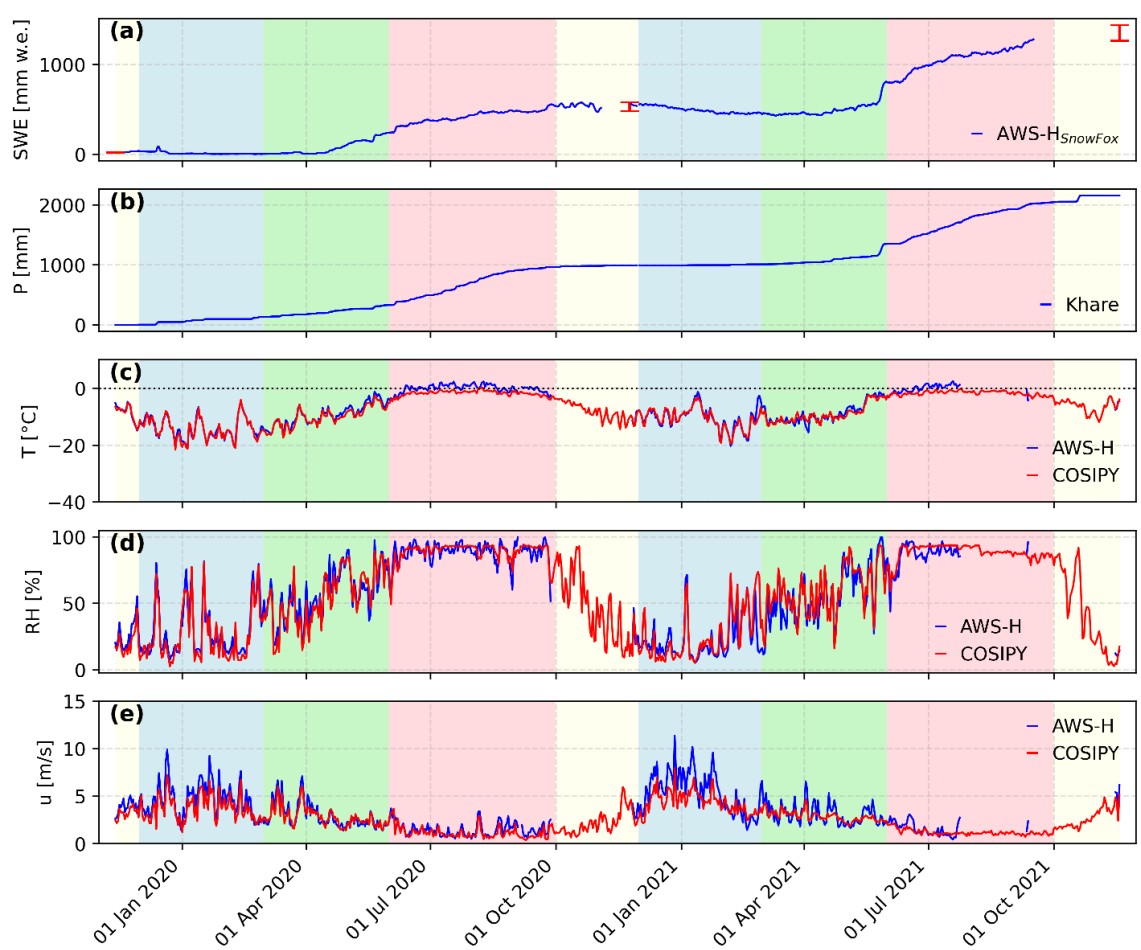

**Figure 2. Daily values of (a) Cumulative SWE measured by SnowFox (blue line) and manual field measurements with error bars (in red), (b) cumulative precipitation at Khare, (c-d-e) air temperature, relative humidity, and wind speed, respectively, measured at AWS-H (blue) and used to force COSIPY at the corresponding grid cell (red) (see Khadka et al. (2024) for details). The black dashed line in panel c corresponds to T = 0°C. Light red, yellow, blue and green shaded areas represent the monsoon, the post-monsoon, the winter and pre-monsoon, respectively.**

Figure 2a clearly identifies a seasonal pattern in snowpack formation, with accumulation during the pre-monsoon and monsoon, typically characterized by regular snowfall, higher air temperatures (Fig. 2c), increasing relative humidity (Fig. 2d), and decreasing wind speeds (Fig. 2e). Between 12 November 2019 and 24 November 2020, the snowpack accumulated 540 mm w.e., while between 24 November 2020 and 18 November 2021, it accumulated slightly more with 742 mm w.e. This

larger total amount in 2021 can be attributed to the occurrence of two typhoons, Tauktae and Yaas, on 18 May and 29 May 2021, respectively, which contributed 215 mm w.e. precipitation at Khare station and 181 mm w.e. of SWE at AWS-H. Conversely, SWE tends to decrease during the post-monsoon and winter due to the low air temperatures, lower humidity, and higher winds driving increased snow sublimation and wind erosion. A spectacular example of such processes was observed on

14 December 2019: a sudden increase of 60 mm w.e. was recorded by the SnowFox, which was also captured by the Geonor gauge in Khare (Fig. 2b), with 44 mm w.e. of precipitation recorded but, within two days following this event, the SWE decreased by 40 mm w.e. due to strong winds. Subsequently, the cumulative SWE monotonically decreased and reached 1 mm w.e. by the end of February 2020.

Differences are detected between (i) the monthly cumulative precipitation at Khare; (ii) changes in SWE at AWS-H measured by the SnowFox; and (iii) the mass fluxes simulated by COSIPY (Fig. 3). During the winter months, the monthly SnowFox SWE changes are typically negative, even when accumulation is observed at Khare, which we attribute to sublimation and erosion due to strong winds (Fig 2e; (Litt et al., 2019). The simulated cumulative sublimation between November and March is as high as 93 mm w.e. and 74 mm w.e. in 2019/20 and 2020/21 respectively (67 % and 70% of annual sublimation), compared to 173 and 54 mm of total precipitation recorded at Khare (only 18 and 6% of annual precipitation). The mass removal due to wind related processes is likely underestimated in COSIPY, as the model accounts only for sublimation and not for wind erosion of snow, which is significant during post-monsoon and winter seasons due to strong high-altitude winds (Brun et al., 2023). In May, June, and September, the increases in SWE measured by the SnowFox are more similar to the Khare precipitation totals.

For the core monsoon months (July-August), the two years of measurements are very different. In the first year (2019/20), the change in SWE is small compared to the precipitation at Khare (100 mm w.e. vs. 384 mm w.e.), while in the second year the values are closer (185 mm w.e. vs. 300 mm w.e.). As the COSIPY simulations show the qualitative importance of refreezing (about 80% of the meltwater), we interpret the difference as a differential refreezing efficiency, because in the second year the SnowFox was buried deeper in the snowpack (323 cm on 18 November 2021) compared to the first year (120 cm on 24 November 2020). Indeed, when the sensor is located closer to the surface (in the first year), a greater proportion of the percolating water is likely to freeze below the sensor and thus not be measured by SnowFox. This interpretation is supported by the systematic observation of ice lenses in the snowpack from the surface to several meters below during field work.

Still our interpretation of the role of refreezing remains very speculative, as it relies on the mass fluxes simulated by COSIPY model. Refreezing is particularly difficult to simulate in snowpack evolution models because it is linked with percolation that is a heavily parametrized process, in particular through a bucket approach in COSIPY (Sauter et al., 2020). Specific experiments, such as temperature profiles, are needed to determine whether meltwater percolates below the previous year's layer, contributing to internal accumulation. The precipitation phase is also a major source of uncertainty in our work. We use the phase separation as a function of temperature which is implemented by default in COSIPY. In our case, it predicts only snowfall at the SnowFox for the study period, but we have no direct observation of the precipitation phase.

A significant source of uncertainty arises in SWE measurements is the sensitivity of the CNRS to change in SWE as the snowpack thickness increases. According to Howat et al. (2018), the standard deviation in SWE rises from less than 0.1 cm at 15 cm SWE to 0.5 cm at 50 cm SWE. Similarly, Gugerli et al. (2019) also reported that their cosmic ray sensor overestimated

SWE by +2% ± 13% compared to manual observations, highlighting the inherent uncertainties in autonomous measurements. This variability can result in substantial overestimations or underestimation of SWE, underscoring the importance of validation against in-situ measurements.

This interpretation has a major implication for field measurements, as it suggests that monsoon meltwater can percolate below the previous year's autumn horizon before freezing. Consequently, point surface mass balance measurements made in the lower part of the accumulation area of Mera Glacier (Wagnon et al., 2021) may miss some of this internal accumulation and are likely to be negatively biased. This problem may be common to summer accumulation type glaciers.

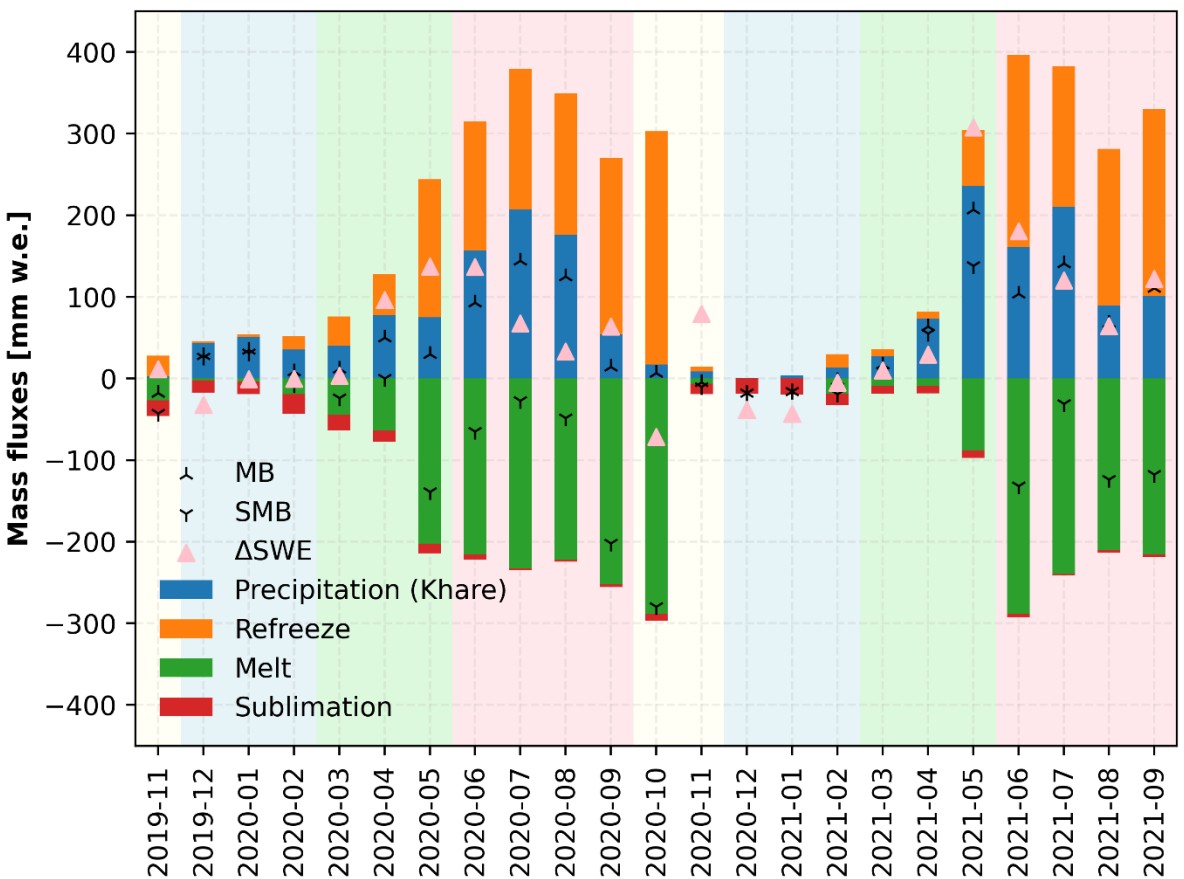

**Figure 3. Monthly values of precipitation at Khare (blue bars), measured changes in SWE at AWS-H (pink triangles) and mass fluxes at the corresponding grid cell simulated by COSIPY: melt (green bars), refreezing (orange bars), sublimation (red bars), surface mass balance SMB (= precipitation + melt + sublimation) (downward black arrow) and mass balance MB (= precipitation + melt + refreezing + sublimation) (upward black arrow). Light red, yellow, blue and green shaded areas represent the monsoon, the post-monsoon, the winter and pre-monsoon, respectively.**

## 5 Conclusions

Operating a cosmic ray sensor (SnowFox) in the accumulation zone of Mera Glacier for two years has allowed continuous monitoring of changes in SWE, which agrees well with limited field observations in November 2020 and 2021. Our analysis highlights seasonal variations in SWE, with accumulation occurring in the pre-monsoon and monsoon, followed by ablation in the post-monsoon and winter due to sublimation and wind erosion. Further investigation of the mass balance components using COSIPY model revealed a significant amount of meltwater percolation and refreezing within the snowpack, explaining how physical processes can contribute to the seasonal evolution of the snowpack. This interpretation has a major implication for field measurements as it suggests that monsoon meltwater can percolate below the previous year's autumn horizon before freezing, but the interpretation remains speculative, as the different mass fluxes are not observed directly by estimated from COSIPY model.

We also stress that, depending the depth of the cosmic ray sensor burial, different processes can dominate: when the sensor is close to the surface, it is likely less sensitive to refreezing than when it is buried deeply. This is a challenge when burring CRS in an existing snowpack, like in firn area of glaciers. We conclude that the expansion of such measurements could provide much improved distributed estimates of snowpack evolution and governing processes, complementing more resource-intensive manual measurements on remote, high altitude Himalayan glaciers.

## 6 Data availability

All data will be deposited in a repository pending publication.

## 7 Acknowledgements

This work has been supported by the French Service d'Observation GLACIOCLIM (part of IR OZCAR). This work would not have been possible without the International Joint Lab Water-Himal (principal investigators D. Shrestha, and P. Wagnon) supported by IRD and all the efforts from people in the field: porters, students, and helpers who are greatly acknowledged here. This research was conducted in partnership with National Geographic Society, Rolex and Tribhuvan University, with approval from all relevant agencies of the Government of Nepal.

## 8 Author contribution

Conceptualization: PW, FB, TM, BP; Data curation: NP, PW, AK; Formal analysis: NP; Investigation: NP, PW, FB, AK, MR, AG; Funding acquisition: PW, TM, BP; Writing (original draft): NP, PW, FB, AK, MR, AG, TM, BP

## 9 Competing interest

The authors declare that they have no conflict of interest.

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
