# Peer review of "Brief Communication: Accurate and autonomous snow water equivalent measurements using a cosmic ray sensor on a Himalayan glacier"

_EGUsphere, 2024_

## Author Comment (AC1)

**Brief Communication: Accurate and autonomous snow water equivalent measurements using a cosmic ray sensor on a Himalayan glacier**

We thank the editor and reviewers for their detailed review of this paper, which has helped us to improve it. We have done our best to address all the comments and suggestions made by the editor and reviewers. Below are the detailed responses to each comment. All comments are in bold black font and our responses are in regular font. Where changes were substantial, we have copied and pasted the sentences from the revised manuscript in italics between quotation marks, with the changed sections in red and the unchanged section in black.

**RC1: The study presents an interesting combination and comparison between SWE measurements by a cosmic ray sensor (CRS), precipitation measurements and model simulations by COSIPY in the lower accumulation area of a Himalayan glacier in Nepal. With their measurement setup, the authors draw important conclusions on processes in the accumulation area of a glacier. The overall results and conclusions are very interesting and worth a publication. However, revisions are needed to provide more clarity on the measurements and the conclusions and to improve the readability of this manuscript.**

**The following open questions/ points should be addressed:**

Thank you for your review of our work, and comments that will help improve the manuscript.

**However, revisions are needed to provide more clarity on the measurements and the conclusions and to improve the readability of this manuscript.**

Many thanks for the comments and suggestions. We have copied all the major and minor comments, and the detailed responses to all the comments are below.

1. **RC1: The title is somewhat misleading. The title gives the impression that the study presents a thorough evaluation of the cosmic ray sensor in these high-altitude areas. Yet, there are only three manual measurements presented as a reference at the same site (of which one has no corresponding CRS observations (see Fig.2). For the title to be less misleading, the word "Application" could be integrated (or something in that sense).**

Thank you for your suggestion. We agree that the original title could be perceived as misleading regarding the scope of the evaluation of the cosmic ray sensor. Based on your suggestion, we have revised the title to better reflect the focus of the study. The new title is "*Application of a Cosmic Ray Sensor for Autonomous Snow Water Equivalent Measurements on a Himalayan Glacier*".

2. **RC1: It is not outlined how the authors dealt with the main issue of deploying a CRS in an accumulation area of a glacier: Is the device dug out every season, or will it slowly be buried in the glacier until it does not work anymore?**

Thank you for your observation. The device was not dug out every season, but after two years. We allowed it to operate continuously until it stopped functioning due to burial of the solar panel.

At that point, we retrieved the device to assess and address the problem. This approach was chosen to maintain the integrity of the data collection over a longer period without frequent interference. This is now clearly written in the data and method section: "*On 12 November 2019, the SnowFox was installed approximately 20 cm below the fresh snow surface. It was progressively buried by snowfalls, until we excavated it on November 2021, roughly two months after it stopped functioning due to burial its solar panel.*"

3. **RC1: What are the absolute neutron count numbers and their evolution over time (maybe add such a plot in the supplement)? What are the uncertainties of the CRS and how strongly do the counts fluctuate over their time intervals?**

Thank you for your suggestion. We have addressed your suggestion by incorporating a time-series plot of absolute neutron count (Figure S1). As we are end-users of the CRS device, we did not investigate the sources of uncertainties on the raw neutron counts, but verified that the retrieved SWE was consistent with observations. The scanning time of the CRS in 20 seconds, that are averaged every 30 min. We record only the average neutron count, and not the standard deviation, so we cannot assess how much the counts fluctuate.

4. **RC1: A Brief Communication should be kept short, but it would still be helpful to dedicate two or three sentences to the COSIPY model given the importance of this model for this study.**

We modified the last paragraph of section 3 to include a more detailed description of COSIPY model. It now reads as:

"*SWE measurements are compared with COSIPY mass flux simulations. COSIPY is a one-dimensional multi-layer python-based model that resolves the energy and mass exchanges of a snowpack/ice column with the atmosphere (Sauter et al., 2020). The COSIPY simulations were produced by Khadka et al. (2024). COSIPY was forced with distributed meteorological forcings originating from an AWS located on the lower part of Mera Glacier (Khadka et al., 2024). We extracted the mass fluxes modelled by COSIPY from the 0.003°×0.003° (0.01 km$^2$) grid cell where the SnowFox and AWS-H are located (Fig. 1).*"

5. **RC1: The authors compare delta SWE derived by CRS measurements to the simulation of mass fluxes by the COSIPY model, but this evaluation is only done qualitatively even though approximately two years of measurements are available. In addition, uncertainties of the CRS measurements (or the COSIPY model) are neither taken into consideration nor discussed.**

We agree that our analysis is rather qualitative. We will add a paragraph in the discussion to highlight the uncertainties related to COSIPY simulations (especially regarding processes like

refreezing in the snowpack). We will also discuss in more details the uncertainties related to the CRS measurements and conversion into SWE, in particular for thick snowpack.

**Detailed comments:**

6. **RC1: L22: Gugerli et al. (2019) present a performance assessment of the CRS and information that can be gained from such measurements. I recommend to use a more suitable references for such a general statement.**

Thank you for your suggestion. The citation has been updated, and the reference now cites Stewart (2009) for the more general statement regarding the importance of seasonal snowpack in high mountain regions.

**"***Seasonal snowpack in high mountain regions is crucial for glaciology, hydrology and climate change research (Stewart, 2009)"*

7. **RC1: L26-27: Please revise this sentence.**

Thank you for pointing this out. The sentence has been revised for clarity and is now as follows:

*"In the logistically challenging environment of the higher Himalayas, accurately measuring precipitation and addressing snow distribution are difficult tasks, which limit the reliability and continuity of SWE measurements (Shea et al., 2015)."*

8. **RC1: L40: Please note that neither Howat et al. (2018) nor Gugerli et al. (2019) measured 4000 mm w.e. above the CRS. Gugerli et al. (2019), e.g., measured approximately 2000 mm w.e. of SWE (and snow depths up to 5m). If this estimate of 4000 mm SWE is taken from Fig. 3 in Gugerli et al. (2019), it corresponds to a theoretical estimate of what the sensor should be able to measure, but this has not yet been demonstrated (at least not to my knowledge).**
   Thank you for pointing out this mistake, and we apologize for the oversight. You are correct that neither Howat et al. (2018) nor Gugerli et al. (2019) measured 4000 mm w.e. above the CRS. We have revised the sentence to reflect the accurate information. The updated sentence now reads:

*"In this study, we deployed a Hydroinnova SnowFox: a cosmic ray sensor (CRS) able to measure SWE up to 2 meters water equivalent or more (Howat et al., 2018; Gugerli et al., 2019) in the lower part of the accumulation zone of Mera Glacier (Central Himalaya, upper Dudh Koshi basin)."*

9. **RC1: L84: How did you define the depth at which to bury it? Did you excavate a pit until the first change of snow density (which is the depth of 20cm you mention)? How was the snow/ firn below that?**

From our previous experience of Mera Glacier accumulation regime, we expected the snow to be mostly eroded by wind after the snowfox being buried (in fall 2019), but we did not want to bury it to deep, because we would need to excavate the next season. The depth of 20 cm was thus a

rather arbitrary choice. The 20 cm depth does not correspond to a change in snow density. The snow of the accumulation area has a density of 380 kg m-3, that corresponds to compacted snow. There is no obvious stratigraphical limit between the snow of the previous year and older snow, due to intense refreezing and wind compaction (Wagnon et al., 2013, 2021; Litt et al., 2019). The revised sentence reads as: "On 12 November 2019, the SnowFox was installed approximately 20 cm below the fresh snow surface, buried at 20 cm to be as shallow as possible, whilst avoiding exposure due to the wind deflation that is commonly observed in the post monsoon and winter."

10. **RC1: L86: Were the counts averaged or summed over the hour? In Howat et al. (2018) and Gugerli et al. (2019), for example, the counts are summed over an hour (cph). It would be interesting to keep the way the counts are presented in the same way to more easily allow for a comparison across different sites.**

In response to your question regarding whether the counts were averaged or summed over the hour: the counts were averaged over the hour. This decision was made because the raw data was obtained in half-hour intervals and applying solar and pressure corrections required using a correction factor that was hourly. Consequently, we used the hourly average to align with the correction methodology.

11. **RC1: L115: It is not very clear how you obtained the reference count. Above you write that the scanning time was 20 seconds, and these counts are averaged over an hour resulting in counts per hour. For the reference count, however, you average 1-minute counts. Wouldn't it be more consistent to use the same measurement strategy in both cases, also to use the same reference period for all the correction factors?**

Thank you for your question, and we apologize for the confusion. We actually noticed a mistake in our manuscript, the scanning time is actually one minute as well. This is now corrected and the sentence (L86 of the original manuscript) reads as: "In this case, the scanning time was one minute"

The reference count was taken in November 2022, after we excavated the SnowFox. We did the same 1-minute scanning interval, but collected the neutron count only for a duration of one hour, due to tough weather conditions. We added a sentence in the revised discussion to stress the limitation of this limited acquisition time.

**RC1: L127: "through good agreement with field measurements" – According to Fig. 2, there are only two field measurements available which can be compared to the SnowFox measurements, and the first one appears to be obtained during the deployment of the SnowFox and hence with a disturbed snowpack.**

Thank you for your question. We acknowledge that the availability of field measurements is limited due to our annual monitoring schedule, which is influenced by weather conditions. Despite having only two field measurements, the agreement between SnowFox measurements and available field data remains remarkably good, given we did not optimize any of equation 6 that relates the neutron count to the SWE. This agreement underscores the value of our

observations and supports the reliability of the SnowFox sensor in capturing snow water equivalent measurements. We believe these limited field observations are still indicative of the sensor's performance and provide valuable insight into its accuracy.

12. **RC1: L130: What does the uncertainty of the manually measured SWE represent - a standard deviation of several measurements, or an error propagation?**

Thank you for your question. The uncertainty of the manually measured SWE represents a standard deviation of several measurements, it is added in the revised manuscript.

13. **RC1: L135: How long was the second gap?**

Thank you for your question. The second gap lasted from September 16, when the sensor stopped working, until November 18, when the sensor was excavated. This results in a gap of 63 days. It is added in the revised manuscript: "137 mm w.e. was recorded during the second gap of 63 days."

14. **RC1: L147: During which time period are these precipitation amounts accumulated? Are they accounted for undercatch (if measured by a gauge)? Occurring during a typhoon, snowfall was probably accompanied by strong winds resulting in significant amounts missed by a gauge observation.**

The precipitation amounts accumulated between May 18 and May 29, 2021, during which two typhoons occurred. Typhoon Tauktae developed over the Arabian Sea on May 13, 2021, and intensified into an extremely severe cyclonic storm by May 17, with its effects detected at the Khare site starting on May 18. Similarly, Typhoon Yaas formed over the Bay of Bengal on May 22, strengthened into a very severe cyclonic storm by May 25, and its influence on the Khare site ended on May 29.

Precipitation during this period was measured using a Geonor T-200B gauge equipped with a wind shield. We apply undercatch corrections are recommended by the world meteorological organization (WMO). More details about these corrections are available in Khadka et al. (2024), and it is now detailed in the manuscript: "Additionally, the Khare Geonor station records precipitation in all weather conditions (Fig. 1b). This Geonor T-200B is equipped with a wind shield and we apply undercatch corrections, as recommended by the World Meteorological Organization (WMO), more details are available in Khadka et al. (2024)."

The mean wind speed during these typhoon events is 1.7 m $s^{-1}$, and the mean air temperature is slightly positive (1.2°C), so we do not expect very large undercatch for these specific events.

Additionally, we apologize for the earlier typo—Typhoon Yaas also occurred in 2021, not 2020.

"*This larger total amount in 2021 can be attributed to the occurrence of two typhoons, Tauktae and Yaas, on 18 May and 29 May 2021, respectively, which contributed 215 mm w.e. precipitation at Khare station and 181 mm w.e. of SWE at AWS-H.*"

15. **RC1: L149:153: Here, it would be nice to have a plot for this period to also see the variations of SWE at the hourly time interval (for example in a supplement). 1 mm SWE obtained by a CRS seems to lie within the natural fluctuations of the CRS.**

We completely agree with your suggestion. In response, we have added a plot (Figure S 1: Change in SWE over time.) that shows the variation of SWE at an hourly interval from December 1, 2019, to February 29, 2020. This should provide a clearer picture of the fluctuations in SWE, including the 1 mm SWE observed by the CRS, which falls within the expected natural fluctuations of the sensor.

[Figure]

16. **RC1: Figure 1: Readability of the integrated table could be improved by aligning the device with the parameter. If someone does not know the measurement devices, it becomes confusing to read SWE and Vaisala in the same line.**

Thank you for your suggestion. In the updated figure, we have improved readability by separating the sensor names and their corresponding parameters with a horizontal line. This should make it clearer and less confusing, especially for those unfamiliar with the measurement

devices.

[Figure]

| Station | Variables (gap % during the study period) | Sensor (uncertainty) |
|---|---|---|
| Khare Geonor, 4888 m a.s.l. Off-glacier, on grassy surface | P (0) | GEONOR T-200BM (± 15%) |
| AWS-H 5770 m a.s.l. On-glacier (accumulation area) | SWE (11) | SnowFox |
| | T (23), RH (23) | Vaisala-HMP45C (±0.2°C; ±2%) |
| | u (23) | Young 05103-5 (±0.3 m/s) |
| | SWin (23), SWout (23), LWin (23), LWout (71) | Kipp & Zonen CNR4 (± 3%) |
| AWS-L, 5360 m a.s.l. On-glacier (ablation area) | T (23), RH (23) | Vaisala-HMP45C (±0.2°C; +2%) |
| | u (25.8) | Young 05103-5 (±0.3 m/s) |
| AWS Mera-La, 5352 m a.s.l. On firm rocks (off-glacier) | $P_a$ (0) | CS100 (±2.0 hPa) |

*"Figure 1. Map of Mera Glacier showing the network of ablation stakes and accumulation pits (pink dots). The stars represent the locations of the AWS used in this study, with in red the location of the SnowFox installed approximately 10 m northwest of AWS-H. The red square represents the grid cell at AWS-H where COSIPY simulations are done. The pictures show the all-weather Geonor precipitation gauge (a), a picture of the SnowFox site showing AWS-H on the right, the SnowFox datalogger mast on the left and the snow trench where the SnowFox is installed (in the front) (b), and the SnowFox during the installation on 12 November 2019 (c). A table (top left) provides detailed information about the stations with variables (temperature (T), precipitation (P), SWE, relative humidity (RH), wind speed (u), atmospheric pressure ($P_a$), outgoing longwave (LWout), incoming longwave (LWin), outgoing shortwave (SWout) and incoming shortwave radiation (SWin)), percentage of data gap for each variable, sensor types and its uncertainty. The outline of Mera Glacier is from 2018 with a total area of 4.84 km$^2$, and the background image was acquired by Sentinel-2 on 24 November 2018. Elevation contours are extracted from the 2012 Pléiades DEM (Wagnon et al., 2021). The inset map gives the location of Dudh Koshi basin (yellow) in Nepal with general pathways of moisture carried by westerlies (left) and Indian summer monsoon (right). Light blue areas are the glacierized areas from Randolph glacier inventory 6 (Pfeffer at al., 2014)."*

**17. RC1: Figure 2: To better follow the descriptions in the text, it would be very helpful to: (i) mark the periods (pre-monsoon, monsoon, etc.) with shadings as is done in Figure 3, and (ii) to better label the dates on the x-axis (e.g., 1 Jan 2020)**

Thank you for the helpful feedback. In response, we have updated Figure 2 as follows: (i) we have shaded the periods (pre-monsoon, monsoon, etc.) like Figure 3 to improve consistency with the descriptions in the text, and (ii) we have enhanced the labeling on the x-axis by including clearer date markers (e.g., 1 Jan 2020) for easier interpretation.

[Figure]

*"Figure 2. Daily values of (a) Cumulative SWE measured by SnowFox (line) and manual field measurements with error bars (red dots), (b) cumulative precipitation at Khare, (c-d-e) air temperature, relative humidity, and wind speed, respectively, measured at AWS-H (blue) and used to force COSIPY at the corresponding grid cell (red) (see Khadka et al. (2024) for details). The black dashed line in panel c corresponds to T = 0°C. Light red, yellow, blue and green shaded areas represent the monsoon, the post-monsoon, the winter and pre-monsoon, respectively."*

**18. RC1: Figure 3: Are the changes in SWE always from the beginning to the end of a calendar month?**

Yes, the changes in SWE are tracked from the beginning to the end of each calendar month, with two exceptions: for the starting month, data collection begins on November 13, 2019, and for the ending month, we only have data up to September 16, 2021. This is now specified in the revised manuscript.

---

## Author Comment (AC2)

**The manuscript presents the use of a point-scale, below snow Cosmic Ray Neutron Sensor (CRNS) for monitoring Snow Water Equivalent (SWE) on a Himalayan glacier. The results are compared to a model run, which is also used to further analyze the hydrological fluxes. While the novelty of the method is minor, as previous research already showed that technique is suitable for glacier monitoring, e.g. on glaciers in Greenland and Switzerland, it constitutes an interesting case study in a data-sparse region. As such, I'd recommend publication after minor revisions.**

Thank you for your kind words and acknowledgement.

**General remarks:**

1. **RC2: I'd suggest to emphasize that access of the region is (presumably) difficult and the region is thus rather (?) data-sparse.**

We will rewrite the first paragraph of the introduction to better highlight the interest of our results in light of limited available measurements. In particular, we will stress the data scarcity in the region and especially at high elevation.

2. **RC2: The section "Results and Discussion" mainly presents results without discussing them. In Particular, I'm missing a discussion on the uncertainties of both the CRNS and the model and their implications as compared to other studies**

Thanks for this comment that echoes main comment 5 of RC1. We agree that our analysis is rather qualitative. We will add a paragraph in the discussion to highlight the uncertainties related to COSIPY simulations (especially regarding processes like refreezing in the snowpack). We will also discuss in more details the uncertainties related to the CRS measurements and conversion into SWE, in particular for thick snowpack. We will also add a comparison to other CRS based studies that were conducted in different meteorological contexts.

3. **RC2: Also, the conclusions could be more elaborated**

We will expend the conclusions to emphasis the implications and limitations of this study. Still we want to keep this section as concise as possible.

**Specific comments:**

4. **RC2: Why is there no air pressure data for the site? It is one of the most important correction factors.**

The automatic weather station (AWS) that was installed alongside SnowFox to capture various meteorological data, AWS-H, included air pressure measurements. Unfortunately, due to an issue with the pressure sensor that might have been damages during transportation, the pressure measurements from this sensor were not reliable. This is the reason why we chose to extrapolate measurements from Mera La AWS located closely, but at a much lower elevation. Due to its vicinity, we expect the hydrostatic assumption to be valid. We added more details in the revised manuscript: "Air pressure is not measured directly at the study site, due to pressure sensor failure"

**5. RC2: The formatting of the citations looks strange, e.g., in line 89/90.**

Thank you for noticing. I have changed the citations correctly.

**6. RC2: L 101/105: I think I understand what you did. But please rewrite this paragraph to make the information more readable as it's quite difficult to follow.**

Thank you for the suggestions. Here is the revised paragraph.

*"Where, K represents a location-specific normalization factor, with a value of 3.08. ε, which equals 1.14, is a correction factor used to adjust the sensitivity of the standard lead neutron monitor. Rc, set at 14.53 GV, refers to the effective vertical rigidity, calculated using the MAGNETOCOSMICS code (part of the Geant4 toolkit, available at crnslab.org). Lastly, χ, equal to 543.51 g cm$^{-2}$, denotes the atmospheric depth, which is derived from local atmospheric pressure (p) and the acceleration due to gravity (g)."*

**7. RC2: L 112/113: How did you exactly derive the attenuation length? It is in a plause range for a cutoff-rigidity of 14.53, but I don't really understand the sentence and the method used here to derive the value from Jungfraujoch data.**

The attenuation length is calculated based on the location and average pressure estimated from elevation. All the computations are done online from the website crnslab.org, and more precisely on the page crnslab.org/util/intensity.php following the method described in McJannet and Desilets (2022). We updated the manuscript to clarify this point:
"The mass attenuation length (L) is taken as 150 g cm$^{-2}$ for our study site, this value was obtained from the online calculator of the crnslab.org, and more specifically using the 'Scaling factor calculator', which is an implementation of Mcjannet and Desilets (2022)."

---

## Author Response (AR2)

Dear Horst,

Thank you for your decision and thank you very much for the last round of comments about the language edits. Please find attached our revised manuscript, as well as a response to your comments (in bold font).

All the best,

Navaraj and co-authors

**Line 88: "...due to burial of its solar panel ..."**

Done

**Lines 106-107: "...with a value of ... neutron monitor." This sentence is unclear. There is a dot after 3.08, should this be a multiplication sign? Probably because of this I do not understand how "3.08.epsilon" equals 1.14. Please also reword the entire sentence: K is first labelled as normalization factor, then as a correction factor? If not both statements refer to K, please make clear what the "correction factor" is.**

We clarified that K and ε are two different factors. The revised sentence reads as: "where K (with a value of 3.08) represents a location-specific normalization factor. ε (with a value of 1.14) represents a correction factor used to adjust the sensitivity of the standard lead neutron monitor."

**Line 147: "over" instead of "overs",**

Corrected

**Line 147-149: please revise the entire sentence, the order of arguments is confusing. Maybe make two sentences.**

We separated the original sentence into two sentences: "During the installation, the sensor was buried beneath about 20 cm of fresh snow. The sensor measured 24 mm water equivalent (w.e.) of SWE, which means that the snow density was approximately 120 kg m$^{-3}$".

**Line 202: "by the COSIPY model"**

Corrected

**Line 210: remove "arises" or change to "arising". Shorter, however, would be better.**

Corrected with "arises" being removed

**Figure 2a: The red dots are not visible. Either draw dots that are larger than the vertical lines of the uncertainty bars or modify description in the caption.**

We modified the caption description: "Daily values of (a) Cumulative SWE measured by SnowFox (blue line) and manual field measurements represented by red vertical error bars"

**Line 241: Please revise the wording.**

We split the sentence. The last parts now read as: "Still this interpretation remains speculative, as the different mass fluxes are not observed directly, but estimated from a surface energy balance model."

**Line 241: "depending on"**

Corrected

**Line 243: "it is likely less sensitive to refreezing": Please explain more clearly. The sensor itself is not measuring refreezing, so it cannot be sensitive to it. I understand what you mean but suggest rewording. e.g. "...when the Snowfox is close to the surface, the measurements are less sensitive to capturing WE of refrozen meltwater..."**

See response below

**Line 244-245: "This is a challenge ..." I do not understand this sentence, please reword.**

We agree that the whole wording of lines 243-245 was not clear. We rephrased the lines 243-245 as: "We stress that there are specific challenges in interpreting cosmic ray sensor measurements when the sensor is located into an existing snowpack (e.g., in the firn area of a glacier). If the sensor is buried close to the surface, meltwater that refreezes deeper than the sensor is not be counted as accumulated SWE. On the contrary, if the sensor is buried deeply, it is more likely that meltwater refreezes above the sensor and is thus counted as accumulated SWE."

**Line 250: Please make sure to include a link to a data repository.**

Done. Note that we also updated the reference to Khadka et al. (2024), which was just published.